# HDHL-INTIMIC: A European Knowledge Platform on Food, Diet, Intestinal Microbiomics, and Human Health

**DOI:** 10.3390/nu14091881

**Published:** 2022-04-29

**Authors:** Valeria Agamennone, Peter M. Abuja, Marijana Basic, Maria De Angelis, André Gessner, Bart Keijser, Martin Larsen, Mariona Pinart, Katharina Nimptsch, Estelle Pujos-Guillot, Kristina Schlicht, Itai Sharon, Eva Untersmayr, Matthias Laudes, Tobias Pischon, Jildau Bouwman

**Affiliations:** 1TNO (Netherlands Organisation for Applied Scientific Research), Microbiology and Systems Biology, 3704 HE Zeist, The Netherlands; valeria.agamennone@tno.nl (V.A.); bart.keijser@tno.nl (B.K.); 2D&R Institute of Pathology, Medical University of Graz, 8010 Graz, Austria; peter.abuja@medunigraz.at; 3Institute for Laboratory Animal Science, Hannover Medical School, 30625 Hannover, Germany; basic.marijana@mh-hannover.de; 4Department of Soil, Plant and Food Sciences, University of Bari Aldo Moro, 70126 Bari, Italy; maria.deangelis@uniba.it; 5Institute of Clinical Microbiology and Hygiene, University Hospital Regensburg, 93053 Regensburg, Germany; andre.gessner@klinik.uni-regensburg.de; 6Centre d’Immunologie et des Maladies Infectieuses (CIMI-Paris), Hôpital Pitié-Salpêtrière, Université Pierre et Marie Curie (UPMC)—Paris 6, INSERM (Institut National de la Santé et de la Recherche Médicale) Sorbonne Universités, 75013 Paris, France; martin.larsen@sorbonne-universite.fr; 7Max-Delbrueck-Center for Molecular Medicine in the Helmholtz Association (MDC), Molecular Epidemiology Research Group, 13125 Berlin, Germany; mariona.pinartgilberga@mdc-berlin.de (M.P.); katharina.nimptsch@mdc-berlin.de (K.N.); 8INRAE, UNH, Metabolism Exploration Platform, MetaboHUB Clermont, Clermont Auvergne University, 63000 Clermont-Ferrand, France; estelle.pujos-guillot@inra.fr; 9Institute of Diabetes and Clinical Metabolic Research, University of Kiel, 24105 Kiel, Germany; kristina.schlicht@uksh.de; 10Migal—Galilee Research Institute, P.O. Box 831, Kiryat Shmona 11016, Israel; itaish@migal.org.il; 11Faculty of Sciences and Technology, Tel-Hai Academic College, Upper Galilee 12210, Israel; 12Institute of Pathophysiology and Allergy Research, Medical University of Vienna, Waehringer Guertel 18-20, 1090 Vienna, Austria; eva.untersmayr@meduniwien.ac.at; 13Max-Delbrueck-Center for Molecular Medicine in the Helmholtz Association (MDC), Biobank Technology Platform, 13125 Berlin, Germany; 14Berlin Institute of Health at Charité—Universitätsmedizin Berlin, Core Facility Biobank, 10178 Berlin, Germany; 15Charité—Universitätsmedizin Berlin, Corporate Member of Freie Universität Berlin and Humboldt-Universität zu Berlin, 10117 Berlin, Germany

**Keywords:** diet, microbiome, health, metabolomics, data sharing, FAIR, data integration, networking, reference microbiome, standardization

## Abstract

Studies indicate that the intestinal microbiota influences general metabolic processes in humans, thereby modulating the risk of chronic diseases such as type 2 diabetes, allergy, cardiovascular disease, and colorectal cancer (CRC). Dietary factors are also directly related to chronic disease risk, and they affect the composition and function of the gut microbiota. Still, detailed knowledge on the relation between diet, the microbiota, and chronic disease risk is limited. The overarching aim of the HDHL-INTIMIC (INtesTInal MICrobiomics) knowledge platform is to foster studies on the microbiota, nutrition, and health by assembling available knowledge of the microbiota and of the other aspects (e.g., food science and metabolomics) that are relevant in the context of microbiome research. The goal is to make this information findable, accessible, interoperable, and reusable (FAIR) to the scientific community, and to share information with the various stakeholders. Through these efforts a network of transnational and multidisciplinary collaboration has emerged, which has contributed to further develop and increase the impact of microbiome research in human health. The roles of microbiota in early infancy, during ageing, and in subclinical and clinically manifested disease are identified as urgent areas of research in this knowledge platform.

## 1. Introduction

During the last 20 years, microbiome research has yielded tremendous insight into the composition of the gut microbiota and identified possible associations of gut microbiota with several nutrition-related diseases. Evidence demonstrates that dietary factors influence the composition and function of the gut microbiota [1,2], and that both diet and intestinal microbiota modulate the risk of several chronic diseases, including type 2 diabetes, allergy, cardiovascular disease (CVD), systemic low grade inflammation (inflammaging), inflammatory bowel disease (IBD), and colorectal cancer (CRC) [3,4]. Still, we lack a good understanding of the relationship between diet, the microbiota, and chronic disease risk.

Knowledge is limited in part by the many confounding factors that influence the association of the gut microbiota with disease, and by the need for much larger study groups in order to determine direct relations between the two. Another limiting factor in microbiome research is the frequent lack of proof of causality. The intestinal microbiota is a highly complex ecosystem [5] and the concept of a healthy resilient gut microbiota relies on its high richness and biodiversity. Lower bacterial diversity has been observed in people with cardiometabolic diseases [6] and irritable bowel syndrome [7] compared with healthy controls, or simply with advanced age, but it is unclear whether these changes precede or follow the onset of disease and aging. Human and rodent intervention studies have addressed these issues using nutritional modifications [8], personalized food and innovative food technology [9], pro- and prebiotics [10], and non-surgical and surgical weight loss procedures [11,12]. However, the comparability of results between studies with different designs, number of participants, choice of control groups, and other factors remains a challenge. A combination of animal models (especially gnotobiotic animals), ex vivo/in vitro and nutritional models, human intervention studies, and “omics” approaches is indispensable to move past association and to demonstrate the cause–effect relationship between the microbiota and the host in health and disease [13,14,15].

The third major challenge encountered by translational microbiome research as well as by any research field, even in large well-established consortia, relates to data integration. It can be challenging to find and access data from microbiome studies. Furthermore, even when data are accessible, there are often difficulties related to its reusability; for example, with regard to comparability across studies. Research outcomes may be affected by different sample collection and processing procedures, thus standardization of methods is key for data comparability. High-throughput sequencing and other omics technologies have shone a light on the large inter-individual variation in microbial community composition and metabolic processes. Methodological bias adds to this complexity, further complicating the development of therapeutic strategies such as dietary intervention or fecal microbiota transfer (FMT). Data integration, achieved through the systematic collection and analysis of available, high-resolution data and through the development of standard operating procedures (SOPs) for sample collection, processing, and storage, and for data analysis and sharing, strengthens translational biomedical research.

The main objective of the INTIMIC knowledge platform is to promote studies on microbiome, nutrition, and health in order to understand the impact of the microbiota on human health. The strategy is focused on assembling available information in the field of microbiome research and in other disciplines that are relevant for nutrition and health (e.g., food science, metabolomics, physiology, and gastroenterology), and on making this information findable, accessible, interoperable, and reusable (FAIR) to the scientific community, the stakeholders, and the general public. A network of transnational and multidisciplinary collaborators is working towards these aims. Three urgent research topics are identified in this knowledge program as a focus area for development: 1. the role of microbiota in early infancy, 2. during ageing and 3. in subclinical and clinically manifest disease (see Table 1 for specific aims). This knowledge program constitutes a crystallization point for the development of a future biomedical sciences research infrastructure.

## 2. Approach

A catalogue of relevant data for the field is being produced for the purpose of data sharing. Many datasets from studies connecting the microbiota, nutrition, and health exist at the European Joint Programming Initiative “A Healthy Diet for a Healthy Life” (JPI HDHL) HDHL-INTIMIC knowledge platform (IKP) partner institutes, within the other JPI HDHL INTIMIC consortia and other places within Europe and beyond. More and more journals require deposition of data as a prerequisite for publication: several databases are available, but a unified system for metadata and annotation is often lacking. Further, European data protection regulations need to be fulfilled when storing human personalized data, which requires, among others, informed consent for storing the data for specific purposes, disclosure about storage place and use of data, and the ability to withdraw consent and to delete personal data. These requirements are usually not fulfilled when personalized human data from ongoing studies would be stored in currently existing public data repositories. The central scientific and technical concept of this JPI HDHL INTIMIC platform is to:Create a catalogue of studies on microbiome, nutrition, and health;Define, for each study type, the vocabulary and metadata variables (e.g., timeline information, participant characteristics, etc.) necessary to minimally define the study setup;Create a catalogue of standard operating procedures for microbiome analyses;Define a reference microbiome for subtypes of subjects;Bring together standards relevant for the community for other (omics) data;Create relevant ontologies where needed.

For data collection, DAta SHaring In Nutrition (DASH-IN), the system developed by ENPADASI (http://www.enpadasi.eu/ access date 20 March 2022), is used. This is a combination of the earlier developed tools Phenotype database (www.dbnp.org, access date 20 March 2022) and OPAL/DataSHIELD [16]. This will result in FAIR data sharing for microbiome–nutrition–health data. We involve other relevant communities (e.g., ELIXIR, BBMRI, metabolomics society, NFDI4Health) to prevent repetitions and to make it possible to link to other FAIR initiatives. For instance, this project is one of the use cases for the Food and Nutrition community in ELIXIR. We are also performing a number of case studies that will highlight the potential of using shared datasets. This facilitates the interest in data sharing by highlighting that sharing normally leads to more scientific output (knowledge, publications, and visibility). Case studies can also help identify solutions for proper attribution of credits when merging shared data deposited in a database. This aspect is of special relevance when omics datasets are used, most of which are largely unexploited by authors in their published studies. Data sharing is also closely connected with the definition of standardized microbiome guidelines, reference microbiomes, and strategies and tools for microbiome research. Finally, all activities are connected to training, dissemination, and networking (see Figure 1).

## 3. Conceptualization

### 3.1. Quality of the Transnational Collaboration and Scientific Exchange

None of the partners or countries has all the knowledge, tools, and/or data that are essential to develop an effective open access platform for microbiota, food, and health. This platform brings together amongst others the data of the previously founded JPI HDHL INTIMIC projects. The collaboration was formed by a national (nine countries) selection of partners. Within nine countries, partner groups were selected due to their wide range of experience in data sharing, nutrition, microbiology, data analysis, ontologies, medicine, metabolomics, and other backgrounds that are relevant for the platform. During the preparatory meeting (in 2018) the selected partners defined the main goals and structure of the platform. This procedure ensured that there would be a direct connection between the expertise in the group, the aims of the current proposal (see Table 1), and the aims in the call. Groups not knowing each other before were stimulated to work together directly, resulting in new capacity building. Within the consortium, those with long-term experience with dissemination of scientific results lead the working groups on this item. 

### 3.2. Data Capture and Sharing

One central aim is the identification of animal and human (intervention and observational) studies from knowledge platform (KP)-INTIMIC partners with a wealth of data relevant to nutrition and the intestinal microbiota. This includes metadata of these studies in order to define ontologies and standardization strategies (WPs 2 and 4) and to define use cases on specific microbiota-related research questions (WPs 6 and 7). Metadata includes, among others, general information on the study (name of study, study web links, funding body, scope, study design, and recruitment), information on measurement of the microbiota (type of samples collected, analysis method, sequencing platform), other laboratory measurements in biological samples (candidate biomarkers, metabolomics, genomics, proteomics), exposure measurements (dietary intake, alcohol and tobacco consumption, physical activity, sedentary behavior, anthropometric measurements, sociodemographic information, and health status), and health-related outcomes. This also includes information on signed informed consents, ethics committee approval, and whether data owners may share raw data or metadata within and outside the KP-INTIMIC consortium.

The INTIMIC knowledge platform offers and expands the DASH-IN infrastructure previously developed in ENPADASI [17], which allows data owners and researchers to share their data and metadata for future research. The DASH-IN infrastructure encompasses two different types of infrastructure: on the one hand the Nutritional Phenotype database (www.dbnp.org , access date 20 March 2022), which is a single central database that facilitates data integration and pooled analysis of individual-level data, and DataSHIELD [16], an open source software project which allows federated joint analyses of individual-level data from several studies conducted without physically transferring their data into a single central database, as a way to circumvent privacy, ethical, and legal constraints of observational studies that often preclude physical sharing of data [18]. Furthermore, the KP-INTIMIC offers a metadatabase based on a Mica server, so that users can interrogate or interpret metadata from the identified studies by means of an initial set of ontologies in agreement with the FAIR principles [17].

### 3.3. Standardization and Guidelines for Microbiome Analysis

The aim of this line of work is to develop a collection of standard operating procedures and tools for the standardization of both wet lab and data analysis procedures. This will be achieved by pooling the expertise of the partners, by collecting information on existing relevant initiatives and by comparing methods in order to determine the overall comparability of microbiome studies. The INTIMIC knowledge platform is creating an inventory of methods leading to a database of standard operating procedures for microbiome sequencing and data analysis. Special attention is paid to methods for analysis of low biomass samples and less frequently used sample materials, as well as comparison of marker-based and whole metagenome shotgun-based sequencing techniques. The primary goal is to identify key aspects for standardization from a broad spectrum of implemented workflows in order to develop guidelines for a quality-controlled implementation of microbiome sequencing into laboratories. Using this data collection, novel control concepts will be developed for the analysis as well as identification and rating of the most critical methodological steps during wet lab processes and data analysis. To further investigate methodological bias and future needs for standardization, schemes for future multi-center ring trials within the consortium are developed. By collecting and rating relevant public standardization initiatives the working group links to important ongoing activities from other projects or consortia (e.g., the Earth Microbiome Project, Critical Assessment of Metagenome Interpretation (CAMI), MicrobiomeSupport).

Further activities in this area:Inventory of relevant initiatives and tools for standardization in microbiome analysis leading to a SOP databank;Analysis appraisal tools;Inventory of relevant initiatives and tools for existing microbiome data integration.

### 3.4. Defining a Reference Microbiome

This line of work focuses on describing the inter-individual variation in the taxonomic and functional composition of the human gut microbiota through the integration of 16S rRNA and metagenomic datasets. The working group takes advantage of the many datasets that have become available in recent years from projects such as MetaHit [19,20], the Human Microbiome Project (HMP) [21], and datasets derived from JPI HDHL projects. The integration of datasets derived using different library preparation procedures is challenging since biases related to these procedures need to be taken into account. As part of its work the working group develops methods for evaluating these biases as well as biases related to the bioinformatics methods that are used. Methods and data that are developed in this working group will be used to compare the microbiota of different populations and age groups and to identify characteristics of the microbiota under specific health conditions and dietary regimes. Moreover, the results of these activities will assist in proposing new standards and guidelines regarding sample processing and sharing. 

### 3.5. Standardization of Other Data and Development of Ontologies

The aim of this working group is to provide a standardized dataset and ontologies to unravel correlations among data originating from different areas of research relative to the axis of functional food, intestinal metabolomics, and human health. Indeed, in connection with the functional food axis and human health, a specific part within this consortium initiative is dedicated to intestinal microbiota-related metabolite and protein analyses. The opportunity to analyze metabolomics and metaproteomics data within interconnected databases from many cohorts allows a comprehensive overview of microbiota composition as well as its downstream production in terms of whole metabolite and protein sets, and therefore its functions. In order to fulfil this objective, the merged data need to be harmonized and the associated measurable variables require an accurate curation.

Regarding 16S rRNA profiles and associated metadata (WP3), customized protocols for term searches and statistical analyses were ad hoc developed and resulted in specific ontologies and standardization protocols that were designed to unravel the existing correlations among inter-individual metabolite and protein variability. Comparison of metabolomics and proteomics data between different studies is still today a challenging task since global approaches result in semi-quantitative data (relative amounts measured), and in various coverage in terms of metabolite/protein diversity. Over a standardization of produced data, a homogenization of data processing, data structure, units, and a nomenclature for naming molecules are required. The whole data science for metabolomics (processing, statistics, annotation) is coordinated and ensured by connection with ELIXIR Metabolomics community and the Metabolomics Society (and its metabolomics standard: MSI). 

The relative databases that were interconnected through the sharing activity in this WP include data on healthy ageing, health perception, behavior (diet, functional food, and physical activity) and metabolomics data, physiological parameters, and other relevant data. The working group activities are divided into four different sub-tasks:Nutritional and functional food composition and ontology;Standardization/ontology building for metadata on physical activity;Standardize metabolomics and other omics data;Microbial fluxes affecting the food–gut microbiota interplay.

The first mentioned topic, fundamental for the exploitation of the others, is the definition of a hierarchical vocabulary, connected through logical relationships of functional food impacting human health. The resulting list of terms defines the ontology and guarantees the groundwork for the proper retrieval of foods belonging to different European cultures, thus including the key elements for the datamining of existing information. 

As reported in recent literature, some cross-sectional evidence proposed physical activity and protein intake as factors impacting richness and diversity of the microbiota in the gut [22]. Thus, the creation of ONS (Ontology for Nutritional Studies) conceptual network in the second sub-task was specifically extended to metadata related to physical activity. Diet and exercise are drivers of biodiversity and occupy a concomitant niche that has been explored.

On another hand, large traditional and fermented foods are also a major topic impacting the gut microbiota. Thus, one of the project sub-task activities is dedicated to harmonizing and standardizing existing data related to foodborne microbiota. In soil–food–gut microbial flow, food can act as vehicle of live environmental microbes between soil and the human gut through plants and animals. Therefore, with the final aim of obtaining a standardized framework connecting diet and health, an important step relies on the collection of available data on the microbiota associated to different foods from different European cultures. A specific database, storing typical foods together with preparation methods, nutritional values, and associated microorganisms (starter and non-starter cultures), provides the link between foodborne (especially dairy products) and variability of the microbiota in terms of taxa composition and metabolite/protein production.

### 3.6. Dissemination, Networking, and Training

The aim of the working group is to coordinate effective network building and dissemination of the knowledge platform (KP) activities, knowledge, resources, and project results. The working group aims to successfully exploit the results of the KP, and to share the acquired knowledge and expertise with the scientific community, the stakeholders, and the general public. This is achieved, among others, through the creation of a community of project partners and relevant stakeholders, and by connecting to the JPI INTIMIC Microbiome projects. On the website of the KP all deliverables of the project are continuously published (https://dashin.eu/jpi-kp, access date 20 March 2022).

The KP brings together more than 60 partner institutions in nine European and associated countries that constitute a substantial share of European microbiome research. The KP can therefore serve as an initial nucleus for a future ESFRI BMS research infrastructure (RI). The most ambitious tasks for developing an RI are assembling the scientific community by cooperation and training, addressing the needs of users and stakeholders, and engaging and informing the general public as the final beneficiary of microbiome research. These objectives are reflected within the tasks of the networking and training work package. 

Assembling the scientific microbiome community is a daunting task since the variation in aims, methodologies, and approaches is enormous. Assembling these within a unique RI requires standardization on several levels, but goes far beyond standardization. It requires interoperation of the various sub-disciplines. The JPI KP may be regarded as a field experiment, in which concepts for the interoperation of these various approaches can be tested and validated. To support this experiment, three thematic working groups were established, each of which performs this exercise in a specific area: (i) early life, (ii) healthy ageing, and (iii) health and disease. Based on these overarching themes, sub-disciplines working closely together in the field of microbiomics are invited to share knowledge, resources, and data to ensure a more uniform training and education level. To achieve this aim, e-learning activities are made available for collaborating institutions. 

The needs of (future) users and stakeholders are equally varied and need to be brought to the table early—ideally in the nucleation phase of an RI. Both users and several groups of stakeholders define the scientific need that justifies establishing an RI; the stakeholders contribute additionally to the sustainability of an RI. Given the complexity of the field, users and stakeholders comprise a broad range of interests, ranging from individual interest in a healthy microbiota, the food and pre/probiotic industry, medicine, and public health services. One specific instrument in serving the users’ and industry stakeholders’ requirements is covered by Expert Centers (EC). ECs were initially developed as a concept during the preparatory phase of the European research infrastructure for biobanks and biomolecular resources (BBMRI-ERIC) [23,24]. ECs serve as trusted intermediary in a public–private partnership and were jointly proposed by biobanks, the industry, and patient organizations [25,26] for the field of biomedical research. Since microbiome research faces similar challenges as ECs, but in a much more diverse way, the adaptation of the EC concept for microbiome research is one task of this KP.

Finally, yet importantly, scientific outreach and involvement of the public is an essential cornerstone of successful dissemination and training. In addition to dissemination of information in an easily understandable language, the role of citizen science is gaining growing interest in research [27]. Microbiomics research is based on big data sets, which can be more easily generated by the involvement of informed patients. Research evidence supports this approach in the field of microbiomics and also highlights the willingness of patients to participate as citizen scientists [28]. 

### 3.7. Human Association Studies

By conducting joint analyses of human study data from various sources identified within the network, the KP-INTIMIC contributes to answer research questions aiming to understand the relationships between the microbiota, diet, and health in the areas of early infancy, ageing, and the development of chronic and sub-chronic diseases. Addressing these questions requires data from observational and intervention human datasets (identified in the data capture and sharing working group), as well as harmonization of data and tools such as common bioinformatics pipelines (such as DADA2, QIIME I, QIIME 2) for standardization of microbiome analysis and data processing developed in other working groups. To accommodate changes in diet and health status throughout life, four case studies are proposed:Diet–Microbiota interactions in early life;Diet–Microbiota interactions in adults;Diet–Microbiota interactions in the elderly;Microbiota in health and disease.

The aim is to provide a structured data resource containing metadata and microbiota data from previously published studies or directly from the knowledge platform partners. The metadata includes dietary patterns as well as other study-dependent clinical information. Finally, we have made particular efforts in reviewing included studies to extract detailed technical information concerning the methods utilized for microbiota analysis, including sample storage, DNA extraction, PCR primers, sequencing technology, and bioinformatical processing.

Research questions are addressed by means of use cases, with a focus on dietary composition in adults and in infants, and on the gut microbiota in health and disease. To address different questions, use cases rely on different methodologies, including use cases based on intervention studies, focusing on acute shorter-term effects, as well as those relying on observational studies, which focus more on longer-term (habitual) exposure.

The effects of specific dietary interventions on metabolic and disease risk factors, may differ between subjects depending on numerous factors, including anthropometry, health parameters, and microbiota composition. Moreover, gut microbiota and its interaction with the host is affected by dietary behaviors and physio-metabolic features. However, currently, knowledge is lacking on which factors may be of importance to explain inter-individual differences in response to specific dietary interventions [29]. Moreover, there are still gaps in understanding how specific diets affect gut microbiota and how this affects health parameters. A large number of dietary intervention studies have been undertaken where data on food intake, microbiota composition, metabolomics, and other health-related biomarkers are available [30,31]. By summarizing and even pooling data this use case aims for full data integration from such studies to create a unique opportunity to strengthen and accelerate the development of personalized nutrition and, at the same time, bring researchers in Europe together in joint actions.

While one of the use cases is focused on dietary composition in adult life, a second use case focuses on dietary composition in early infancy. Breastfeeding versus formula feeding in infants pre-weaning significantly influences early-life gut microbiota colonization. Indeed breastfeeding favors certain microbes, such as *Lactobacillus* and *Bifidobacterium*. Supplementing formula with pre- or probiotics has been proposed to make gut colonization following formula feeding more similar to breastfeeding. Some interventional studies in humans have been conducted in this respect [32,33,34] in addition to studies with child cohorts from farmers where raw milk was used as part of the daily nutrition. 

Protocols of these interventional studies are compared and merged to provide guidelines and ensure efficacy of future studies. Since data on intervention studies is often limited and inconsistent, datasets are enriched with publicly available data from various databases. 

Inflammatory and metabolic factors play an important role in the development of chronic diseases [35]. Some dietary and nutritional factors are related to inflammatory and metabolic factors and associated diseases, and they have been suggested to modulate the composition and function of the gut microbiota [1,2,36,37]. For example, a number of studies have examined the impact of obesity on the microbiota; however, results of these studies have been inconsistent, and different methodologies have been used [38]. In addition, it is unclear whether any potential differences in microbiota composition between obese and non-obese individuals can statistically be accounted for by differences in dietary behavior. The use cases therefore investigate differences in the microbiome composition between individuals with and without a specific health condition or disease. In addition, the association between microbiota composition and inflammation and metabolic factors as well as the modulation capacity of dietary patterns on the microbiota is examined. Finally, the extent to which different types of aging (i.e., healthy aging, aging in nursery homes, aging with disease, etc.) affect the gut microbiota in relation to diseases as well as potential associations between body mass index, physical activity, dietary consumption (fat, protein, carbohydrates, and fiber), and gut microbial diversity across age groups are assessed [39].

### 3.8. Functional Studies in Model Organisms

The aim of this working group is to assemble resources to facilitate and support functional microbiome studies in model organisms by combining expertise in “omics” approaches, animal studies, gnotobiology, and nutrition as well as disease and cell-based models. The working group provides an inventory of available models (in vitro, nutritional, and animal models), samples, and isolates, and shares these inventories with the consortium through a network platform. Additionally, this group assesses current model strategies used to evaluate the causal role of the microbiota on the maintenance of host health and development of pathologies. This working group is responsible for the following activities: Inventory of available models, specimens, and protocols—within this task the partners build an “off-site” biobank and an online database of available models, samples, and isolates available among the knowledge platform partners that can be shared and utilized for the microbiome functional and mechanistic studies.Functional readouts of health—within this activity the partners deliver an opinion paper addressing benefits, drawbacks, and limitations of all relevant animal models and related strategies used to evaluate the causal role of microbiota on host immunity and metabolism.

## 4. Future Perspectives

The platform ensures a solid basis of data and standards that help advance the scientific field on the interaction between the microbiota, nutrition, and health. This knowledge may lead to the development of healthier food and functional food products. This initiative consolidates (open access) nutritional mechanistic, intervention, and epidemiological studies on the relation between the microbiota, nutrition, and health. Standardization is crucial for this development as combining studies depends on the comparability of their data and design. The work on standard operating procedures will have a large impact on future data sharing also outside of the IKP consortium, by linking to initiatives such as ELIXIR. Building a reference microbiome also makes it possible to interpret the outcomes of microbiome studies in an integrated way. 

This work creates a system that allows generalized data sharing and a much better use of nutritional study data for future analyses across data sets and study types. Datasets may be used by lawmakers, medical doctors, and funding institutions. Our project grants intelligent access and categorization of big data (e.g., microbiome, metabolomics), including negative results and unpublished information, which is often unstructured and poorly accessible. The potential outcomes will be easier access to medically and nutritionally relevant datasets, and exploitation of both published and unreleased datasets to the benefit of the wider EU-based scientific community. 

In the longer term, this platform will contribute to formulate better and more informed hypotheses for future studies, and, in some cases, make it possible to answer questions without having to conduct additional studies. The availability of a European infrastructure for nutrition research will enhance the capacity of biologists/nutritionists and clinicians to carry out high-impact studies. The use cases show that new research is possible with already collected data, and how to move from correlation to causation. The platform constitutes an atlas of findings in the microbiota–nutrition–health area that will be useful to improve the health system. This possibility is increased by the presence in the consortium of clinicians that help build the infrastructure and make it suitable to support translational medical research. 

## Figures and Tables

**Figure 1 nutrients-14-01881-f001:**
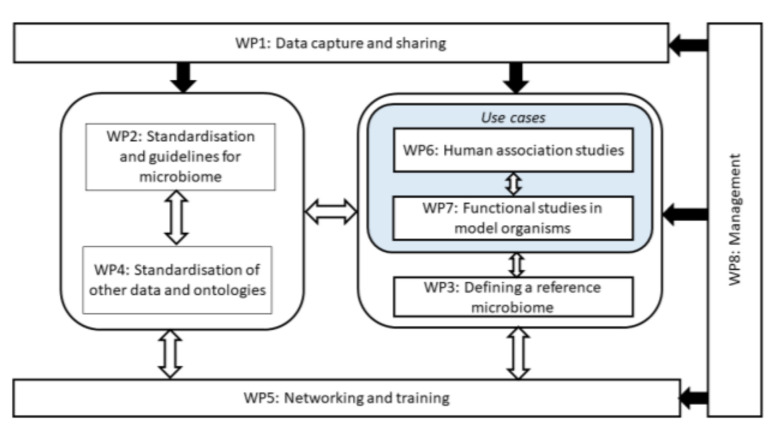
Representation of the various tasks and their interconnection within the INTIMIC knowledge platform. The project is built around use cases both in humans (WP6) and model organisms (WP7). Key in the development on the platform is the definition of a reference microbiome which is done in WP3. Important for the use case is the development of standards, which is done in WP2 and WP4. In order to perform, the use case’s rich (meta) data should be collected in a interoperable way; this is done in WP1. Key to the success of the project is project management (WP8), but especially building the network of researchers within the platform and also the communication of the results to the outside world (WP5).

**Table 1 nutrients-14-01881-t001:** Specific aims of the knowledge platform.

Aim	Description
Data capture and sharing	Identify and collect human and non-human studies with data relevant to food, diet, and intestinal microbiomics, and provide infrastructures to share this data
Standardization and guidelines for microbiome analysis	Provide an overview of the SOPs and tools used within the microbiome field for both wet lab procedures and data analysis, and indications to determine the comparability of outcomes of microbiome studies
Defining a reference microbiome	Describe the intra- and inter-individual variation of the healthy human gut microbiota at the taxonomic and functional levels, by integrating data from 16S rRNA and metagenomic studies
Standardization of other data and development of ontologies	Provide an overview of the standardized datasets and ontologies developed for (omics) analysis platforms, including metabolomics, dietary intake, and physical activity
Networking and training	Promote networking in order to effectively disseminate the activities, knowledge, and resources produced by the program with project partners, external stakeholders, and the public sector
Human association studies	Answer research questions on the relation between the microbiota, diet, and health in the areas of early life and of the development of chronic disease during lifespan and ageing, by using data from observational and interventional human studies and standardization tools developed in the program
Functional studies in model organisms	Provide an inventory of available models (in vitro, nutritional, and animal models), resources, and (multidisciplinary) strategies that can be applied to analyze and validate cause–effect mechanisms in microbiome research; share protocols and SOPs in order to increase reproducibility and comparability

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
