# Peer review of "HDHL-INTIMIC: A European Knowledge Platform on Food, Diet, Intestinal Microbiomics, and Human Health"

_nutrients, 2022, doi:10.3390/nu14091881_

Round 1
Reviewer 1 Report
Agamenonne e col. describe in the manuscript: HDHL-INTIMIC: knowledge platform on food, diet, intestinal microbiomics and human health, a project to create a new integrated platform to better understand the interelations between intestinal microbiota, diet and health. This sort of approach is much needed.Author Response
We would like to thank the reviewer for the positive response.
Response to Reviewer 1 Comments
We have found no comments.
Reviewer 2 Report
I find this article very interesting and valuable. I have minor comments:
I could not find the explanation for the Figure and Table
Line 105 Adds: European Joint Programming Initiative “A Healthy Diet for a Healthy Life” (JPI HDHL)
About Figure 1, It would be very helpful if the authors are able to increase the resolution in any future submission. Please, add more information (description) in Figure 1 footnote
Line 159 Adds: Knowledge Platform (KP)
Line 378 uses italics for Lactobacillus and Bifidobacterium
About the Diet, is it studied by its components (fat, protein, sugar, fiber) or as an eating pattern?
Have you considered studying the eating patterns of the nursing mother?
Author Response
We would like to thank the reviewer for the positive response and the great suggestions.
Response to Reviewer 2 Comments
Point1: I could not find the explanation for the Figure and Table
Response1: I have added references to the figure and table in the main text.
Point2: Line 105 Adds: European Joint Programming Initiative “A Healthy Diet for a Healthy Life” (JPI HDHL)
Response2: We have added this text
Point3: About Figure 1, It would be very helpful if the authors are able to increase the resolution in any future submission. Please, add more information (description) in Figure 1 footnote
Response3: The figure now has a better resolution and a more extensive description of figure 1 has been added.
Point4: Line 159 Adds: Knowledge Platform (KP)
Response4: We have added this text
Point5: Line 378 uses italics for Lactobacillus and Bifidobacterium
Response5: These words are now in italics
Point6: About the Diet, is it studied by its components (fat, protein, sugar, fiber) or as an eating pattern?
Response6: In the platform several Use cases are included that all have their own methodology, which also means that diet can be taken into account differently in the different Use cases. Also the available data will steer the Use cases. As can been seen in this paper of the platform (https://www.ncbi.nlm.nih.gov/pmc/articles/PMC8466729/) most of the collected studies use FFQs, which can be used to study eating patterns. In general it is more difficult to consider eating patterns if comparing studies from different countries than looking at compositions. For this reason there is more focus towards this method.
Point7: Have you considered studying the eating patterns of the nursing mother?
Response7: This is a very important point to consider. Unfortunately for the Use cases in our platform we did not have enough data to include this. We think more studies should collect this key data in order to include this important driver in the future.
Reviewer 3 Report
The manuscript by Agamennone et al, entitled "HDL-INTIMIC: Knowledge platform on food, diet, gut micronomics and human health" is an important and interesting collaborative platform for the European Community to achieve high quality research on factors modifying the gut microbiota, in particular diet, to determine the concept of healthy microbiota, as well as to determine the consequences of gut microbiota modifications, not only in associative terms, but addressing the direct cause-effect of dietary components, in particular in infants, the elderly and in chronic diseases. The platform can accelerate new discoveries through the expertise of collaborators in different aspects associated with the gut microbiota using omics tools, in particular metabolomics and proteomics. This is an excellent initiative, but the title of the article should include that it is a platform for the European Community. Let us hope that some initiative like this will one day also benefit some underdeveloped countries, which are greatly affected by modifications of the gut microbiota, but which do not have the capacity to carry out this type of collaboration.
Author Response
We would like to thank the reviewer for the positive response.
Response to Reviewer 3 Comments
Point1: This is an excellent initiative, but the title of the article should include that it is a platform for the European Community.
Response1: We have added European to the title.